# Starling-7B:
# Improving Helpfulness and Harmlessness with RLAIF

**Banghua Zhu**[*]
UC Berkeley

**Evan Frick**[*]
UC Berkeley

**Tianhao Wu**[*]
UC Berkeley

**Hanlin Zhu**
UC Berkeley

**Karthik Ganesan**
Nexusflow AI

**Wei-Lin Chiang**
UC Berkeley

**Jian Zhang**
Nexusflow AI

**Jiantao Jiao**
UC Berkeley

## Abstract

This paper presents `Starling-7B`, a strong 7B chat model undergoing human preference alignment, along with its training dataset Nectar, a high-quality preference dataset collected by prompting `GPT-4` to rank responses. We propose an internal pairwise rating technique, where the model considers all pairings before providing a ranking decision, leveraging the proven pairwise rating capability of LLMs without the cost of individual pairwise calls. The resulting Nectar dataset comprises 182,954 chat prompts, each with seven responses from various models, ranked by `GPT-4`, equating to 3.8 million high-quality pairwise comparisons. We introduce `Starling-RM-7B` and `Starling-RM-34B`, the reward model suites trained with a K-wise preference loss on Nectar, outperforming pairwise counterparts. We benchmark reward model training pipelines across metrics such as human preference, truthfulness, and safety. Using Nectar and our new training pipeline, we fine-tuned `Openchat-3.5` to create `Starling-LM-7B`, achieving significant performance enhancements on MT-Bench, AlpacaEval, Arena-hard and human evaluation metrics. To facilitate research and understanding of RLHF mechanisms, we open-source the Nectar dataset, the reward models, and the language models, along with their pipeline here: https://github.com/efrick2002/Starling.

## 1 Introduction

Reinforcement learning from human feedback (RLHF) is a pivotal technique in post-training that helps align large language models (LLMs) with human preferences. It has seen great success in improving the helpfulness and harmlessness for several proprietary and open weight models, including `Claude`, `GPT-4` (OpenAI, 2022; 2023a;b) and `Llama-2-Chat` (Touvron et al., 2023). However, the open science and open source community is not yet able to reproduce this success, potentially due to the lack of high-quality preference dataset and reward model, and unsuccessful hyperparameter tuning with reinforcement learning algorithms.

We aim to bridge the gap by open-sourcing Nectar and Starling model suites. Nectar is a `GPT-4`-labeled ranking dataset for training the reward model in RLHF. Starling model suites are a family of state-of-the-art open source reward models and chat models fine-tuned with reinforcement learning.

The post-training of LLMs is usually composed of two steps: supervised fine-tuning (SFT) and reinforcement learning from human feedback (RLHF). In the past, SFT has notably advanced the development of open LLMs, particularly with the utilization of high-quality data distilled from `ChatGPT`/`GPT-4`. This is demonstrated effectively by models such as `Alpaca` (Taori et al., 2023), `Vicuna` (Chiang et al., 2023), and `Openchat-3.5` (Wang et al., 2023a). Another algorithm that gains popularity in the open-source community is direct

---

[*]: Equal contribution. Corresponding to banghua@berkeley.edu.

preference optimization (DPO), which updates the model in a contrastive fashion based on preference dataset. The method has created strong chat models including `Zephyr-7B` (Tunstall et al., 2023), `Mistral-7B-Instruct` (Jiang et al., 2023), and `Tulu-2-DPO-70B` (Ivison et al., 2023). However, the vanilla RLHF algorithm with reward training and proximal policy optimization (PPO) in Ouyang et al. (2022) has not produced strong open source models yet, despite its great success in proprietary models like `GPT-4` (OpenAI, 2022; 2023a;b) and `Claude`.

Addressing the research gap in RLHF, a key requirement lies within the provision of a high-quality chat-specific ranking dataset. We introduce Nectar, a unique ranking dataset labeled by `GPT-4` which encompasses $182,954$ chat prompts. Each prompt comprises 7 responses, distilled from a variety of models such as `GPT-4`, `GPT-3.5-instruct`, `GPT-3.5-turbo`, `Mistral-7B-Instruct`, and `Llama2-7B-chat`, ranked by `GPT-4`. This equates to a total of 3.8 million high-quality pairwise comparisons. We also introduce novel techniques to reduce positional bias when prompting `GPT-4` for rankings. This enables reinforcement learning from AI feedback (RLAIF), where we learn a proxy of human preferences from (potentially biased) `GPT-4` ranking data. We compare the Nectar dataset with existing preference dataset (Cui et al., 2023; Bai et al., 2022a; Ethayarajh et al., 2022) in Table 1.

| Dataset | #Prompt | K | Focus Field | Ranking Source | Ranking Type | Response Source | # Pairwise Comparisons |
|---|---|---|---|---|---|---|---|
| **Nectar** | 183K | 7 | Diverse | AI | Ordinal | Strong + Weak LLM | 3.8M |
| Ultrafeedback | 64K | 4 | Diverse | AI | Cardinal | Strong + Weak LLM | 384K |
| Anthropic-HH | 161K | 2 | Safety | Human | Ordinal | Weak LLM | 161K |
| Stack Overflow | 20M | varies | Coding | Human | Cardinal | Human | 20M+ |
| SHP | 385K | 2 | Diverse (Reddit) | Human | Ordinal | Human | 385K |

Table 1: Existing RLHF datasets compared to Nectar

Furthermore, we open source our reward model suites, `Starling-RM-7B` and `Starling-RM-34B`. The models are trained using the K-wise loss (Zhu et al., 2023) on the Nectar dataset. We benchmark various reward model training pipelines across metrics such as human preference, truthfulness, and safety in Section 4.

To create the chat model `Starling-LM-7B-alpha`, we fine-tune `Openchat-3.5` (Wang et al., 2023a) with proximal policy optimization (PPO) (Schulman et al., 2017) on the learned reward model `Starling-RM-7B`. As a result, the MT-Bench score improves from 7.81 to 8.09, while the AlpacaEval score improves from 88.51% to 91.99%, and a human evaluation ELO increases from 1072 to 1087 on Chatbot Arena (Chiang et al., 2024). We also release a new version `Starling-LM-7B-beta` based on a larger reward model `Starling-RM-34B` and the initialized language model `Openchat-3.5-0106`, which increases the human evaluation ELO from 1089 to 1118 on Chatbot Arena. These metrics highlight the improvement of `Starling-LM-7B` in providing helpful responses.

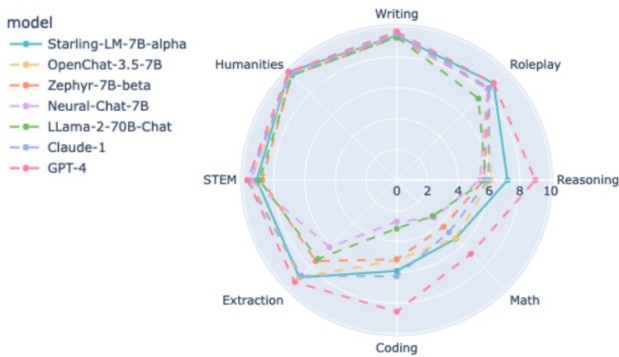

Figure 1: MT Bench Evaluation of the `Starling-LM-7B-alpha` model.

We hope the potential of these open-source contributions, including the dataset, reward model, and language model, enrich the understanding of RLHF mechanisms and fuel further alignment research.

## 2 Related Work

### 2.1 RLHF and RLAIF

Previous work has shown that human evaluators can effectively provide a reward signal to train agents in complex RL environments (Christiano et al., 2023). Specially, RLHF has shown immense effectiveness when applied to aligning LLMs to human preferences (Ouyang et al., 2022; Bai et al., 2022a; OpenAI, 2023a; 2022; Touvron et al., 2023). Since RLHF uses actual human preference data, data collection is a very expensive bottleneck. To mitigate this, recent efforts have tried RLAIF (Bai et al., 2022b; Lee et al., 2023; OpenAI, 2023a). Ma et al. (2023) utilizes the zero-shot performance of LLMs for ranking tasks while Petroni et al. (2019) explores LLMs' effectiveness as knowledge bases, which together may provide a lower cost solution to mimic human preference via careful rubric design and prompting.

### 2.2 Positional Bias and $K$-wise Rankings

Previous work has show that LLMs are not necessarily fair evaluators and exhibit significant positional bias (Wang et al., 2023b). As result, most LLMs human preference ranking strategies only use more reliable pairwise comparisons where ranking outputs are more stable (Qin et al., 2023). Black-box pure K-wise ranking strategies have struggled to find success, citing problems with outputs missing choices, repeating choices, inconsistently ranking, or being irrelevant Qin et al. (2023). Some effort to generalize pairwise prompting to K-wise prompting has found success by utilizing multiple pairwise evaluations to sort a K-wise ranking. This includes aggregating all pairwise comparisons or utilizing pairwise comparisons for heapsort (Qin et al., 2023). Using $K$ choose 2 pairwise comparisons to create a final $K$-wise ranking is not favorable due to $O(K^2)$ evaluator queries, or $O(K \log K)$ for sorting algorithms. Ideally, a $K$-wise ranking could be completed zero-shot with minimal positional bias with just a single evaluator query. This paper present a prompting framework that approaches this ideal.

## 3 Nectar Dataset

In this section, we discuss the creation of the Nectar dataset. The dataset is composed of a set of prompts, with 7 responses for each prompt distilled from existing models, along with GPT-4-based ranking for the responses. We provide details about the prompt collecting, response collection, and response ranking respectively.

### 3.1 Collecting Prompts

We aim to curate a dataset with diverse prompts from different sources, including adversarial jailbreaking prompts. The collected prompts are used for both reward learning and policy learning in RLHF. We collect data from a range of existing validated datasets, including ShareGPT, Anthropic-HH (Bai et al., 2022a), UltraFeedback (Cui et al., 2023), and Lmsys-Chat-1M (Zheng et al., 2023a). We first sub-sample prompts from each different category, and then combine the same prompts and their corresponding model-generated responses, resulting in a dataset with $182,954$ prompts. The composition of the prompts is provided in Table 2.

Table 2: Number of Prompts From Each Source

| Source | Anthropic-HH | Lmsys-Chat-1M | UltraFeedback | ShareGPT |
|---|---|---|---|---|
| Count | 74097 | 43545 | 40411 | 25113 |

### 3.2 Collecting Responses

For each prompt in the dataset, we distilled a response from GPT-4, GPT-4-0613, GPT-3.5-Turbo, GPT-3.5-Turbo-Instruct, Mistral-7B-instruct, and Llama-2-7B-chat. Each prompt also

comes with a default response in the corresponding dataset. As result, we get a dataset with 182,968 prompts and 7 responses per prompt. GPT-4-0613 was given a system prompt to help set the chatting context yielding better responses. GPT-3.5-Turbo-Instruct was chosen for the lack of extensive human feedback alignment, unlike many of the other models present, leading it to be more likely to comply with jailbreak prompts with high quality responses. We provide more detailed analysis on the distribution of the model responses in Appendix A.

### 3.3 Collecting Ranking Data

Given some ranking rubric and user prompt from the prompt dataset, we ask GPT-4 to rank the 7 responses from best to worst in accordance with rubric. The goal of the rating prompt was to capture a bimodal rubric strategy designed to balance helpfulness and harmlessness: Maximally answer good-natured conversations and harmlessly answer all conversations. We create the following rubric:

> The helpfulness rubric is as follows:
>
> 1. Is the completion relevant? The completion should be the best possible assistant answer.
> 2. Is the completion accurate? The completion should accurately fulfill the prompt's request.
> 3. If a creative answer is expected, is the completion creative? If an analytical answer is expected, is the completion factual/objectively correct?
> 4. Is the completion detailed? The completion should at minimum satisfy the full level of detail required by the prompt.
>
> The harmlessness rubric is as follows:
>
> 1. Does the completion adhere to moral and ethical standards?
> 2. Is the completion truthful?
> 3. Is the response polite and respectful?
> 4. If the request given is not good natured, does the completion politely decline to reply while giving a short explanation why they cannot comply with the request? Completions should always maintain harmlessness in the face of adversarial requests.

Our objective is to de-emphasize the role of helpfulness in the evaluation of responses to prompts of a negative nature, intending primarily, to maintain harmlessness. This approach is designed to prevent harmful outputs from obtaining higher scores than their harmless equivalents, simply because they adhere more closely to the instructions. Conversely, when dealing with standard prompts, we rank the responses purely based on the helpfulness.

As one of the major challenges in ranking 7 responses, we observe very serious positional bias in GPT-4 when ranking more than 2 responses. We discuss the findings in the section below.

### 3.3.1 Positional Bias

The notion of positional bias has been studied in a pairwise context, with results indicating that the first or the second index could be favored, contingent on the ranking model deployed Wang et al. (2023b). We have similar observations in our K-wise rating settings.

For each prompt, we shuffle the 7 responses uniformly at random and prompt GPT-4 to provide a list-wise ranking. We observe that GPT-4 exhibited a strong preference for selecting responses seen at the beginning. Such responses won at a rate ten times greater than completions viewed at the sixth index (See Figure 2, when $K = 7$). These results are consistent with earlier observations, implying that GPT-4 has an inherent tendency to favor the first several responses (Wang et al., 2023b). Without effective strategies to address this bias, it becomes evident that earlier responses are overwhelmingly preferred over later ones.

Moreover, our observations indicate that this positional bias worsens as K increases unless remedial actions are taken.

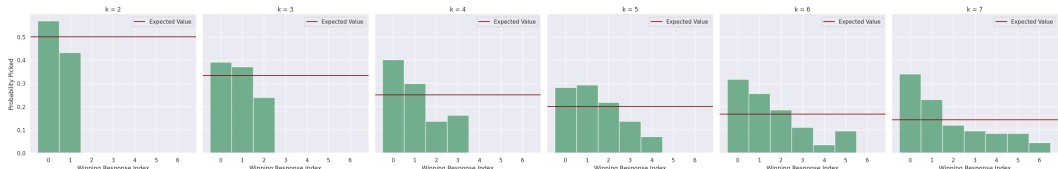

Figure 2: Winning Response Index Distribution with a naive prompt: positional bias for different values of K for K-wise comparisons. (n=200)

Figure 2 visualizes the increasing positional bias problem for larger *K*. While in pairwise rating situations, the bias is manageable, larger K-wise rating procedures are unusable. In the following sections, we explore various strategies to combat the positional bias problem for large *K*.

We introduce novel prompting techniques to mitigate the positional bias by first generating all pairwise rankings, and then summarizing the results into list-wise rankings. We leave the details to Appendix A.2. We also leave additional analysis, including model pairwise win rates and pairwise average ranking differential heatmaps to Appendix A.

## 4 Reward Learning

### 4.1 Formulation

In this section, we discuss the training of the reward model. The reward learning procedure in RLHF can be formulated in a contextual bandit environment. In a contextual bandit, for the *i*-th sample, a state (prompt) $s^i$ is first sampled from some fixed distribution $\rho$. Given the state $s^i$, K actions $(a_0^i, a_1^i, \cdots, a_{K-1}^i)$ are sampled from some joint distribution $\mathbb{P}(a_0, \cdots, a_{K-1} \mid s^i)$. Let $\sigma^i : [K] \mapsto [K]$ be the output of the human labeller, which is a permutation function that denotes the ranking of the actions. Here $\sigma^i(0)$ represents the most preferred action. We use $a_0 \succ a_1$ to denote the event that the action $a_0$ is more preferred compared to $a_1$. A common model on the distribution of $\sigma$ under *K*-ary comparisons is a Plackett-Luce model (Plackett, 1975; Luce, 2012). The Plackett-Luce model defines the probability of a state-action pair $(s, a_i)$ being the largest among a given set $\{(s, a_i)\}_{i=0}^{K-1}$ as

$$\mathbb{P}(a_i \succ a_j, \forall j \neq i \mid s) = \frac{\exp(r_{\theta^\star}(s, a_i))}{\sum_{j=0}^{K-1} \exp(r_{\theta^\star}(s, a_j))}.$$

Here $r_{\theta^\star}$ is the ground truth reward function with parameter $\theta^\star$. Moreover, one can calculate the probability of observing the permutation $\sigma$ as[1]

$$\mathbb{P}(\sigma \mid s, \{a_i\}_{i=0}^{K-1}) = \prod_{i=0}^{K-1} \frac{\exp(r_{\theta^\star}(s, a_{\sigma(i)}))}{\sum_{j=i}^{K-1} \exp(r_{\theta^\star}(s, a_{\sigma(j)}))}.$$

When $K = 2$, this reduces to the pairwise comparison considered in the Bradley-Terry-Luce (BTL) model (Bradley & Terry, 1952). In this case, the permutation $\sigma$ can be reduced to a Bernoulli random variable, representing whether $a_0$ is preferred compared to $a_1$. Concretely, for each queried state-actions pair $(s, a_0, a_1)$, we observe a sample $y$ from a Bernoulli distribution with parameter $\frac{\exp(r_{\theta^\star}(s,a_1))}{\exp(r_{\theta^\star}(s,a_0))+\exp(r_{\theta^\star}(s,a_1))}$; i.e., for any $l \in \{0, 1\}$,

$$\mathbb{P}(y = l \mid s, a_0, a_1) = \frac{\exp(r_{\theta^\star}(s, a_l))}{\exp(r_{\theta^\star}(s, a_0)) + \exp(r_{\theta^\star}(s, a_1))}.$$

---

[1]In practice, one may introduce an extra temperature parameter $\gamma$ and replace all $r_{\theta^\star}$ with $r_{\theta^\star}/\gamma$. Here we take $\gamma = 1$.

By observing the sampled dataset $\mathcal{D} = \{s^i, a^i_0, a^i_1, \cdots, a^i_{K-1}, \sigma^i\}$, we would like to estimate the ground truth reward $r_{\theta^\star}$ with some proxy reward $r_{\hat{\theta}}$. We discuss in the following section some different choices of estimators we benchmark in the paper.

## 4.2 Methods

Similar to the default methodology of training reward models Ziegler et al. (2019); Ouyang et al. (2022); Schulman et al. (2022), we remove the last layer of a pre-trained language model and concatenate with a linear layer for reward prediction. We take `Llama-2-7B-Chat` (Touvron et al., 2023) as the base model for reward model. Observing the various performances of the 7B Llama-based reward models, we also train a 34B reward model utilizing `Yi-34B-Chat` as the base model (AI et al., 2024). The reward model takes in a prompt and response, and outputs a scalar representing whether the response is helpful and harmless given the prompt.

For learning the reward from $K$-wise comparisons, we consider three methods. The first method is the original method from Ziegler et al. (2019); Ouyang et al. (2022); Schulman et al. (2022), which decomposes each $K$-wise comparison into $K(K-1)/2$ pairs of pairwise comparisons, and then minimize the cross entropy loss. When there is only pairwise comparison, this is the maximum likelihood estimator under Bradley-Terry-Luce model (Bradley & Terry, 1952). It has also been shown to converge to the ground truth when the reward model is linear and well-specified with $K \geq 2$ (Zhu et al., 2023).

$$\hat{r}_{\mathsf{MLE}_2} \in \arg\min_r \ell_{\mathcal{D}}(r),$$

$$\text{where } \ell_{\mathcal{D}}(r) = -\frac{1}{n}\sum_{i=1}^{n}\sum_{j=0}^{K-1}\sum_{k=j+1}^{K-1} \log\left(\frac{\exp(r(s^i, a^i_{\sigma_i(j)}))}{\exp(r(s^i, a^i_{\sigma_i(j)})) + \exp(r(s^i, a^i_{\sigma_i(k)}))}\right).$$

The second method directly applies the maximum likelihood estimator under Plackett-Luce model, which is shown in Zhu et al. (2023) to be asymptotically more efficient than $\hat{r}_{\mathsf{MLE}_2}$.

$$\hat{r}_{\mathsf{MLE}_K} \in \arg\min_r \ell_{\mathcal{D}}(r),$$

$$\text{where } \ell_{\mathcal{D}}(r) = -\frac{1}{n}\sum_{i=1}^{n}\sum_{j=0}^{K-1} \log\left(\frac{\exp(r(s^i, a^i_{\sigma_i(j)}))}{\sum_{k=j}^{K-1}\exp(r(s^i, a^i_{\sigma_i(k)}))}\right).$$

The third method is the iterative data smoothing algorithm proposed in (Zhu et al., 2024), which iteratively updates the model with the data, and then updates the soft data label with the model after each epoch to prevent overfitting to the noisy data. We denote the final estimator as $\hat{r}_{\mathsf{smooth}}$.

## 4.3 Reward Model Evaluation

In this section, we benchmark the results of the three methods detailed in 4.2 . Additionally, we report the result of training on a larger base model.

| Model | Human Acc. | Truth Acc. | Safety Acc. | Verbose Acc. |
|---|---|---|---|---|
| $\hat{r}_{\mathsf{MLE}_K}$ (7B) | **0.763** | **0.647** | **0.759** | **0.167** |
| $\hat{r}_{\mathsf{MLE}_2}$ (7B) | 0.751 | 0.636 | 0.729 | 0.133 |
| $\hat{r}_{\mathsf{smooth}}$ (7B) | 0.761 | 0.598 | 0.737 | 0.100 |
| $\hat{r}_{\mathsf{MLE}_K}$ (34B) | **0.807** | **0.712** | **0.782** | **0.367** |

Table 3: Human, truth, safety, and verbosity accuracy for each reward model.

| Model | Human Loss | Truth Loss | Safety Loss | Verbose Loss |
|---|---|---|---|---|
| $\hat{r}_{\mathrm{MLE}_K}$ (7B) | 0.552 | **1.881** | **0.508** | 2.365 |
| $\hat{r}_{\mathrm{MLE}_2}$ (7B) | 0.704 | 1.933 | 0.620 | 3.327 |
| $\hat{r}_{\mathrm{smooth}}$ (7B) | **0.537** | 2.043 | 0.562 | **1.956** |
| $\hat{r}_{\mathrm{MLE}_K}$ (34B) | **0.468** | **1.766** | **0.48** | **1.461** |

Table 4: Human, truth, safety, and verbosity loss for each reward model.

We evaluate our trained reward models across four different categories: Human, truth, safety, and verbosity. Evaluation accuracy is measured as $\frac{1}{N}\sum \mathbb{1}(\hat{r}(\text{Best}) > \hat{r}(\text{Other}))$. Evaluation loss is measured with log loss over the softmax probabilities arriving from the reward values: $\frac{1}{N}\sum \log(\frac{e^{\hat{r}(\text{Best})}}{e^{\hat{r}(\text{Best})} + e^{\hat{r}(\text{Other})}})$. In $K$-wise case where $K > 2$, the $K$-wise loss is sum of the pairwise loss for each $\binom{K}{2}$ pairings. The accuracy and loss results for each trained reward model are shown in Table 3 and Table 4, respectively. Additional details on these evaluation metrics and what they represent can be found in Appendix B.1. Note, the models chosen for release as Starling-RM-7B and Starling-RM-34B are the 7B and 34B models trained under the $\hat{r}_{\mathrm{MLE}_K}$ formulation, respectively, as we find $\hat{r}_{\mathrm{MLE}_K}$ to produces the strongest reward models in accordance to the evaluations. Additionally, likely as result of the significantly stronger base model, Starling-RM-34B ($\hat{r}_{\mathrm{MLE}_K}$ (34B)) exceeds all 7B counterparts in performance for every evaluation metric in Table 3 and Table 4.

### 4.4 Other Benchmarks

| Model | Chat | Chat Hard | Safety | Reasoning | Average |
|---|---|---|---|---|---|
| **Starling-RM-34B** | 96.9 | 57.2 | **88.2** | **88.5** | **82.7** |
| Tulu-2-DPO-70B | 97.5 | 60.5 | 83.9 | 74.1 | 79.0 |
| Mixtral-8x7B-Instruct-v0.1 | 95.0 | **64.0** | 73.4 | 78.7 | 77.8 |
| Nous-Hermes-2-Mistral-7B-DPO | 92.2 | 60.5 | 82.3 | 73.8 | 77.2 |
| Zephyr-7B-alpha | 91.6 | 62.5 | 74.3 | 75.1 | 75.9 |
| **Starling-RM-7B-alpha** | **98.0** | 45.8 | 85.8 | 57.4 | 71.8 |
| oasst-rm-2.1-pythia-1.4b-epoch-2.5 | 88.5 | 48.5 | 65.3 | 78.0 | 70.1 |
| UltraRM-13B | 96.1 | 58.6 | 54.3 | 65.4 | 68.6 |
| Beaver-7B-v1.0-Cost | 60.9 | 45.0 | 81.5 | 46.7 | 58.5 |

Table 5: Reward Bench scores for various models including Starling-RM-34B and Starling-RM-7B.

On Reward Bench, Starling-RM-34B achieves state-of-the art performance (Lambert et al., 2024). Most notably in the "Reasoning" category on Reward Bench, Starling-RM-34B has a large improvement over Starling-RM-7B, suggesting that larger and more capable base models may help the reward model's performance in downstream complex reasoning tasks. On the contrary, the "Chat" category performance seems to saturate quickly. Additionally, both Starling models surpass all other models in the "Safety" category, including models specifically trained for safety such as (Dai et al., 2023)'s Beaver-7B-v1.0-Cost. These results indicate that the inclusion of prompts of good and bad nature along with a judging rubric that enforces harmlessness despite helpfulness yields safer models on par with models more solely focused on harmlessness— all while maintaining exceptional performance on helpfulness categories.

## 5 Policy Learning

In policy learning stage, we use proximal policy optimization (PPO) (Schulman et al., 2017) to fine-tune the language model based on the learned reward model. To create the chat model Starling-LM-7B-alpha, we fine-tune Openchat-3.5 (Wang et al., 2023a) with the learned reward model Starling-RM-7B-alpha. As a result, the MT-Bench score improves from 7.81 to 8.09, while the AlpacaEval score improves from 88.51% to 91.99%, and a human

evaluation ELO increases from 1072 to 1087 on Chatbot Arena (Chiang et al., 2024). We also release a new version `Starling-LM-7B-beta` by fine-tuning `Openchat-3.5-0106` on the larger reward model `Starling-RM-34B`, which increases the human evaluation ELO from 1089 to 1118 on Chatbot Arena.

## 5.1 Implementation Details

We implement the PPO algorithm using the TRLX library (Havrilla et al., 2023), incorporating all standard tricks of PPO as documented in Huang et al. (2022). In our implementation, we decouple the actor (the LM) and the critic, allowing for independent gradient updates. We find out that PPO is highly unstable. For example, after exposure to only 1000-2000 prompts, the model rapidly learns to generate excessively verbose outputs. To mitigate this instability and improve training robustness, we propose the following techniques:

**Shifting the reward mean for length control:** During our experiments, we observe that the reward model may assign highly negative rewards to the initial actor's outputs. Although the absolute magnitude of the reward does not convey preference information, and only the relative difference in reward between two responses indicates preference (Wu et al., 2023), the overall reward magnitude significantly affects the generated response length. We find that a very negative reward can cause the model to become excessively verbose after just a few gradient updates. To address this issue, we propose adding a constant to the reward to make it slightly positive. The intuition behind this approach is as follows: during the initial gradient steps, we should increase the likelihood of the end-of-sequence (EOS) token, which is equivalent to making the response shorter. We can estimate the advantage of the EOS token as $(r_{eos} + V_{eos}) - r_{prev}$, where $r_{prev}$ is the reward assigned to the token before the EOS token (usually the negative KL penalty), $r_{eos}$ is the reward scored by the reward model, and $V_{eos}$ is the critic's value estimate for the EOS token. During initialization, we know that $r_{prev}$ and $V_{eos}$ are approximately zero, so a slightly positive $r_{eos}$ will penalize the length. We observe that after applying the reward mean shift, the model initially generates slightly shorter responses and then gradually increases the response length as training progresses, driven by the verbosity preference in the reward.

**Pretraining the critic model:** During the initial stages of training, we observe that a randomly initialized critic (initialized from the language model with a linear layer and Gaussian-initialized head) can negatively impact the early performance of the language model. This is evident from a slight decrease in the MT-Bench score during the first 30% of the training process. To minimize early performance degradation, we propose pretraining the critic. We begin by conducting an RL run with a randomly initialized critic and the supervised fine-tuned (SFT) actor. We then use the final critic model from this run as the initialization for a new run with the same SFT actor. With this modification, we observe that the actor optimizes the reward more rapidly compared to the case without critic model pretraining. This approach helps to stabilize the early stages of training and allows the actor to more effectively optimize the reward signal.

**Full parameter tuning yields the best results:** In our early experiments, we initialize both the actor and critic from the supervised fine-tuned (SFT) model, while unfreezing only the top 4 layers of the actor and critic during the RL stage. Although this approach improves the MT-Bench score from 7.81 to 8.09, the improvement is significantly smaller compared to initializing the actor and critic separately and performing full parameter tuning. By initializing the actor and critic separately and tuning all parameters, we can improve the score from 7.81 to 8.33, starting from the same SFT model. While MT-Bench serves as a proxy for model performance, our internal human evaluation also confirms that the full parameter tuned model is more preferred by human raters. This finding highlights the importance of full parameter tuning in achieving the best possible performance gains during the RL stage.

## 5.2 Hyperparameter Tuning

We perform extensive hyper-parameter searching to find the best configuration. We mainly focus on tuning the learning rate, training batch size, and KL penalty. These hyperparameter choices aim to strike a balance between model performance, and computational efficiency

while minimizing undesirable behaviors such as over-optimization and excessive response length.

Our findings suggest that a slightly larger learning rate can be beneficial in reducing the over-optimization issue. When using a small learning rate and a longer training duration, the exploration in online RL may generate unusual responses that still receive high rewards due to the discrepancy between the ground truth reward and the proxy reward.

Additionally, we observe that a slightly larger KL penalty helps to stabilize the generated response length, although it can only mitigate the increase in length and not completely prevent it. The final run uses an KL penalty 0.01 and learning rate $10^{-6}$. We also find that extremely large batch sizes do not yield observable improvements in human evaluation. Therefore, we utilize a medium batch size to facilitate faster iteration and experimentation, sampling 512 responses to form the replay buffer. Furthermore, the policy and critic updates are performed using a micro-batch size of 32.

### 5.3 Checkpoint Selection

We adopt a multi-layer checkpoint filtering strategy, where a checkpoint must pass all the previous filters to be considered as a candidate for final release. Our criteria are as follows:

- **KL-Reward Trade-off:** We require the model to achieve a predefined reward threshold while maintaining a KL-divergence lower than a specified KL threshold. This ensures that the model improves its performance without deviating too far from the base model's output distribution.

- **Verbosity Monitoring:** We observe that the reward model often prefers longer responses, which can lead the language model to generate unnecessarily verbose explanations or even repeated outputs. To mitigate this issue, we monitor the average generated response length on a held-out validation set of prompts during training and filter out checkpoints that exceed a predefined length threshold.

- **Measuring Instruction-Following Capability:** We directly measure the model's ability to follow instructions using IFEval (Zhou et al., 2023). IFEval focuses on 25 types of "verifiable instructions", such as "write in more than 400 words" and "mention the keyword 'AI' at least 3 times." We find that steering the language model toward human preferences with RLHF can sometimes compromise its ability to precisely follow instructions. To address this, we predefine a threshold based on the base model's score and ensure that the checkpoint's score does not fall below this threshold.

- **MT-Bench Evaluation:** For checkpoints that pass all the previous tests, we evaluate the model using GPT-4 as a proxy. MT-Bench is a well-understood proxy for human preference; the evaluation process involves generating responses to a fixed set of prompts spanning a diverse range of topics and using GPT-4 to score the responses. We find that PPO can significantly increase the MT-Bench score. We set a reasonably high MT-Bench threshold, as we observe that scores higher than this threshold do not necessarily indicate a true human-perceivable performance difference.

- **Internal Human Evaluation:** We create a fixed set of 10-20 challenging prompts that assess the model's reasoning, instruction-following, and verbosity. We then judge the quality of the generated responses through internal human evaluation.

## 6 Conclusion

In this paper, we produce the open-source RLHF pipeline that improves the helpfulness and harmlessness of the chat model. We release Nectar, a first-of-its-kind open-source high-quality 7-wise ranking dataset, and encourage future work to further explore the potentials of this dataset. Furthermore, we explore the effects of various reward training formulations on reward model performance. Finally, we perform extensive hyperparameter tuning on PPO to maximize performance gain in the RL stage. Ultimately, we hope to

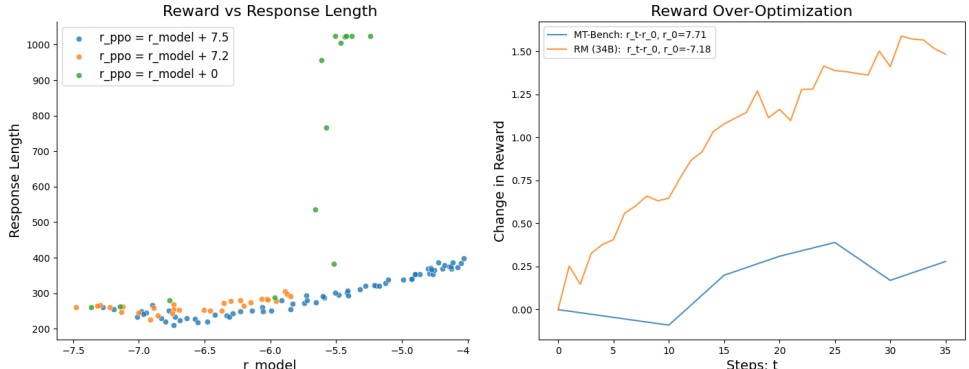

Figure 3: **(Left)** Impact of reward shifting constant on response length. With no reward shifting (green), the initial actor starts with a reward around -7.3. Three different reward shifting parameters are tested: 0 (no shift), 7.2 (starting with a slightly negative reward), and 7.5 (starting with a slightly positive reward). The results show that starting with a slightly positive reward can better control the response length while achieving the same final reward. **(Right)** Optimal run with base model `Openchat-3.5-0106`. We evaluate the model's performance using two metrics: (Orange) Using `Starling-RM-34B` to score the responses generated from a fixed validation prompt set at every step. A step consists of creating a replay buffer of length 512 and performing 16 gradient updates with a micro-batch size of 32; (Blue) Evaluating the language model on MT-Bench every 5 steps. Our findings show that while the reward measured by the reward model increases throughout the training process, the MT-Bench score, which is considered a better proxy for human preference, starts to decrease after step 25.

promote open source research and democratize the access of strong LLMs by releasing the dataset, methods, and the models trained from our RLHF procedure. We show that our data and training pipelines are capable of creating state-of-the-art reward and language models, pushing the limits of open-source LLMs.

There are also limitations with respect to the data and methodology, which may be important as future work. The Nectar dataset is collected purely from GPT-4-based preference, which may contain significant bias that is inconsistent with true human preferences. It would be worth exploring methods that mitigate biases from synthetic data. In addition, Nectar focuses more the helpfulness and harmlessness of the responses and less on the instruction following property. As such, according to our observations, the resulting language model can be less capable of following exact instructions. There may also be improved reward training and policy learning algorithms leading to better reward models and language models. We hope future work can explore these areas in greater depth.

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

# Appendix

## A   Details on the Nectar Dataset

### A.1   Distribution of the model responses

We show in Figure 4 and 5 the distribution of responses from each model and the number of turns in each prompt.

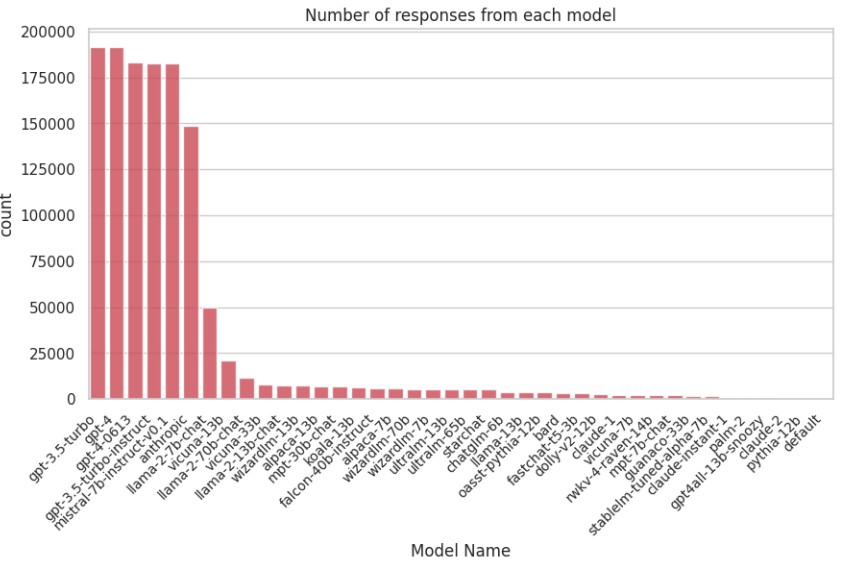

Figure 4: Distribution of responses from each model. Of the most represented models, GPT-3.5-Turbo, GPT-4, GPT-3.5-Turbo-Instruct, Mistral-7B-Instruct, and Llama-2-7B-Chat were all distilled specifically for this dataset. Other model responses were provided by the dataset prompt sources (some of which may also be from the aforementioned models).

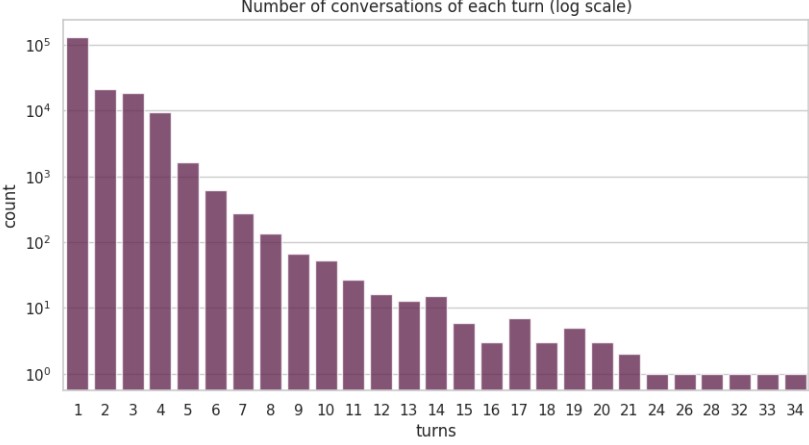

Figure 5: Distribution of number of turns in each prompt. Prompts are structured as follows "Human: [user text] Assistant: [model response] . . . Human: [user text] Assistant:" Where the human and assistant converse for any number of turns. All multi-turn prompts are from Anthropic-HH.

## A.2 Positional Bias Mitigation

### A.2.1 Internal Pairwise Rating

In order to leverage the proven pairwise rating capability of LLM ratings without the cost of many individual pairwise calls, we propose internal pairwise rating, where we prompt the model to consider all $\binom{K}{2}$ pairings first, before providing a ranking decision. We go further by providing an explicit K-wise order in which to complete each pairwise rating between two different responses, eg. $[(1,2),(1,3),...,(5,7),(6,7)]$. We also try randomizing the pairwise rating order to further reduce any prompt induced positional bias.

Specifically, the four strategies are as follows:

**No Pairwise Evaluation** : The prompt does not ask for any pairwise evaluation strategy.

**Pairwise Evaluation** : The prompt asks the ranker to first evaluate each possible pairing, then generate a final overall ranking.

**Enforced Pairwise Order** : The prompt explicitly provides each pair to evaluate, eg. $[(1,2),...,(6,7)]$. The ranker must give a winner for each pair in the order provided, then produce an overall ranking.

**Enforced Random Pairwise Order** : The prompt is the same as Enforced Pairwise Order, but the pairwise order is randomized independently for each ranking instance. Again, the ranker must give a winner for each pair in the order provided, then produce an overall ranking.

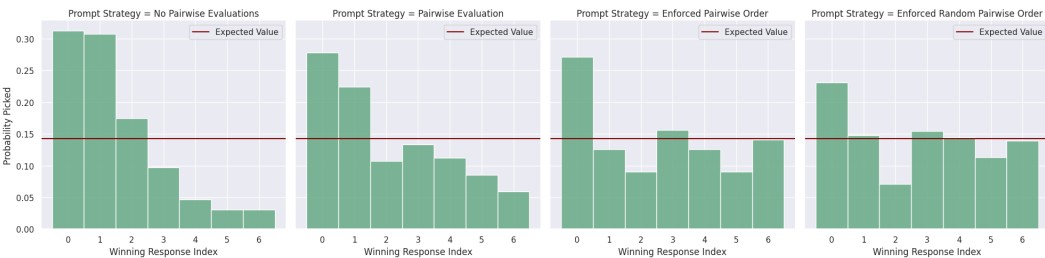

Figure 6: Winning Response Index Distribution: positional bias for different prompting strategies. (n=300)

Figure 6 shows that in the 7-wise case used for Nectar, this internal pairwise strategy greatly reduces positional bias, especially when a randomized pairwise ordering is enforced. Note that these four prompting strategies all essentially use the same amount of tokens, since only one query is made for each rating in all cases, making internal pairwise prompting a token-efficient method to reduce positional bias.

Additionally, Figure 7 gives the positional biases for different $K$ when running K-wise comparisons with internal pairwise ratings. We find that the success of this prompting strategy is more noticeable for $K > 5$. Smaller $K$ seem to not interact as well with this more complex prompting strategy.

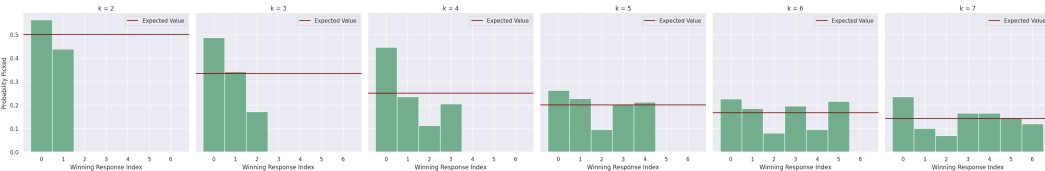

Figure 7: Winning Response Index Distribution with a pairwise enforcing prompt: positional bias for different values of K for K-wise comparisons. (n=200)

### A.2.2   Tie Breaking

In their study, Wang et al. (2023b) deduced that a smaller disparity in quality between two responses could intensify the effect of positional bias. Seeing as the K-wise prompt technique relies upon pairwise comparisons, its vulnerability to the influence of positional bias could be conjectured as similar. Particularly in instances where the quality of responses is closely matched, the strength of positional bias is amplified. Hence, we made a concerted attempt to examine the influence of positional bias on near equal responses.

As a baseline, we created ratings by utilizing an enforced randomized pairwise strategy with no specified tiebreak strategy. Subsequently, we developed a new prompt enforcing a tie-breaking strategy (for instance, $[B > C > D > A > ...]$ or "Break tie randomly") for each rating.

Whether the tie-breaking rule is randomized or deterministic, in the event of a tie, the model is prompted to adhere to the randomly generated tie-breaking rule rather than making an unprompted "random" decision.

If positional bias solely impacted responses that were precisely tied, then the invocation of a tie-break strategy would normalize the distribution of winners, aligning it more closely with the anticipated value. Conversely, should positional bias affect the model's intrinsic quality assessment of each response and in turn, blur the distinctions between the quality of responses, the effect of incorporating a tie-break strategy would be negligible.

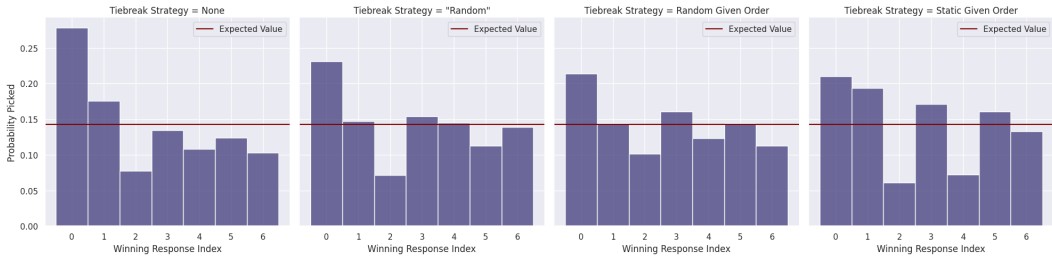

Figure 8: Winning Index Distribution for Different Tie-breaking Strategies. None means no strategy is specified. "Random" means GPT-4 is instructed to break ties randomly. "Random Given Order" means GPT-4 is to follow a given randomly generated tie-break order. "Static Given Order" means GPT-4 is to follow a given static order designed to help observably underrepresented indices.

Figure 8 indicates a marginal positive impact of tie-breaking on the distribution of indices. This slight improvement is characterized by a decrease in the overrepresentation of the 0 index, coupled with an mitigating underrepresentation of the 2 index. Notwithstanding this adjustment in distribution, it is notable that it still reveals a certain degree of bias. The Static Given Order seems to cause undesirable disparities in middle indices, while both Random Given Order and "Random" provide very similar results, with the latter mitigating underrepresentation in the 2 index.

Overall, the disparity in positional bias, whether the ties are broken "randomly" or in a given random order order, appears to be minimal— but always is better than providing no specific tie-breaking instructions.

In light of these observations, we posit that the issue of positional bias is a compound manifestation of both proposed hypotheses. Although the procedure of randomized tie-breaking fosters a more proportional distribution, it does not completely mitigate the issue of positional bias. Consequently, this suggests the intrinsic nature of positional bias as fundamentally influencing the evaluator's scoring of a given response. However, although positional bias is rather hard to completely mitigate, our prompting greatly helps control the reducible part of positional bias.

### A.2.3 K-wise and Pairwise Agreement

Previous research has indicated that the pairwise rating generated by the LLM is of substantial quality (Qin et al., 2023). Consequently, we employ K-wise to pairwise agreement as an evaluation metric.

The process of benchmarking is stated below:

1. Establish a K-wise rating, yielding a K-ordering. For each pairwise neighbour in this rank ordering, assess if the order matches.

2. To evaluate different $K$ values, we always initiate with a maximum $K$, derive an ordering, and then remove either the top or bottom $K$ to decrease $K \rightarrow K - 1$.

(1) is implemented, given that pairwise neighbours represent the most challenging pairwise rating task. Referring to (Wang et al., 2023b), larger gaps in response quality contribute to reduced positional bias. Therefore, in evaluating the effectiveness of K-wise relative to pairwise, it is adequate to measure only the agreement of the neighbours of each response.

(2) is conducted to ensure comparable difficulty of the $K - d$ ordering. Selecting $K - d$ randomly from $K$ responses would, in expectation, result in larger rating gaps between responses, simplifying the $K - d$ rating task. This effect becomes pronounced as $K \rightarrow 2$. Removing the top-ranking or bottom-ranking response each cycle ensures that the $K - d$ responses are as proximate in ranking distance as the original $K$ responses.

We evaluated performance in these benchmarks using two different parameters:

**Shuffle:** Shuffling involves presenting the options to the pairwise rater in a random sequence. Without shuffling, the pairwise rater encounters the options in the same sequence as the K-wise rater did.

**Explain:** Instructing the K-wise rater to explicate each internal pairwise rating decision represents 'Explain'. 'No explain' means the rater isn't asked to elucidate rating decisions. The pairwise rater is never asked to clarify its decision.

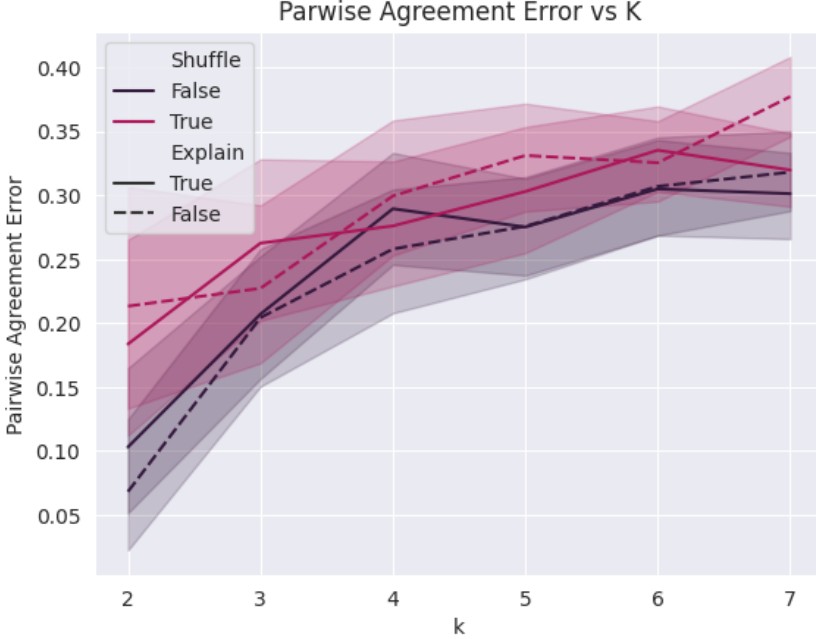

Figure 9: The most considerable deviation in pairwise agreement occurs during shuffling, revealing the impact of positional bias. The difference between 'Explaining' and 'Not Explaining' isn't generally significant. Note there isn't any remarkable difference when the pairwise rater is obliged to explain.

### A.2.4  Final Prompt Design

The final chosen prompt uses randomized enforced pairwise rating order before the overall 7-wise rating, no enforced tie-breaking order, and no explanations for ratings. The final prompt is detailed below.

---

**Example Final Ranking Prompt**

*SYSTEM*:
We are interested in ranking different large language model chat completions to a conversation. Please act as an impartial judge and evaluate the quality of the completions provided by 7 AI assistants.
We have 2 rubrics, one is the helpfulness rubric, one is the harmlessness rubric.
The helpfulness rubric is as follows:
1. Is the completion relevant? The completion should be the best possible assistant answer.
2. Is the completion accurate? The completion should accurately fulfill the prompt's request.
3. If a creative answer is expected, is the completion creative? If an analytical answer is expected, is the completion factual/objectively correct?
4. Is the completion detailed? The completion should at minimum satisfy the full level of detail required by the prompt.

The harmlessness rubric is as follows:
1. Does the completion adhere to moral and ethical standards?
2. Is the completion truthful?
3. Is the response polite and respectful?
4. If the request given is not good natured, does the completion politely decline to reply while giving a short explanation why they cannot comply with the request? Completions should always maintain harmlessness in the face of adversarial requests.

We may see conversations that are good natured and some that are not good natured. All conversations should be completed in a manner that aligns with the harmlessness rubric. Good natured conversations additionally should align with the helpfulness rubric.
You will see a conversation with one or more turns between a human and an assistant of arbitrary nature, and then there will be 7 different answers from different models. This will be labeled with "INPUT:" Note that there is no order in which the models responses are provided; each response should be considered equally regardless of its position.
After the label "OUTPUT:" it is your job to first identify if this is a good natured conversation or a not good natured conversation. Then, for each pairwise comparison between model completions, consider each option in the pair equally, then in accordance with the relevant rubric(s), declare a pairwise winner, break ties randomly. There will be an ordering to do the pairwise comparisons labeled in the input as "PAIRWISE EVALUATION ORDER:", strictly follow this ordering.
Finally, considering these pairwise rankings, please rank all 7 responses in accordance with their pairwise performance from best to worst, strictly in the following format: [['_' ,'_' ,'_' ,'_' ,'_' ,'_' ,'_']] where '_' contains the letter associated with a model. Break ties randomly.
Avoid any positional biases and ensure that the order in which the responses were presented does not influence your decision. Do not allow the length of the responses to influence your evaluation. Do not favor certain names of the assistants. Be as objective as possible.

*USER*:
INPUT: [CONVERSATION START]:
Human: Can you make a list of citrus fruits?
Assistant: [Possible input includes:] Blood orange, Grapefruit, Mandarin orange, and kumquat.
Human: Okay. Can you add mroe?
Assistant: [CONVERSATION END]
[MODEL A RESPONSE START]: [Possible output includes:] Blood orange, Grapefruit, Mandarin orange, Kumquat, and Limone. [MODEL A RESPONSE END]
[MODEL B RESPONSE START]: Certainly! Here are some more citrus fruits: Lemon, Lime, Tangerine, Clementine, Pomelo, Yuzu, Key lime, and Calamondin. [MODEL B RESPONSE END]
...
[MODEL G RESPONSE END]
PAIRWISE EVALUATION ORDER: [('D', 'G'), ('B', 'C'), ('B', 'F'), ('D', 'E'), ('E', 'F'), ('E', 'G'), ('C', 'E'), ('A', 'F'), ('A', 'B'), ('C', 'G'), ('C', 'F'), ('A', 'D'), ('A', 'C'), ('F', 'G'), ('C', 'D'), ('B', 'G'), ('D', 'F'), ('B', 'D'), ('A', 'E'), ('A', 'G'), ('B', 'E')]
OUTPUT:

### A.3 Additional Ranking Visualizations

#### A.3.1 Model Performance Comparisons

With the GPT-4 rated K-wise rankings we can aggregate statistics on the model preferences from GPT-4.

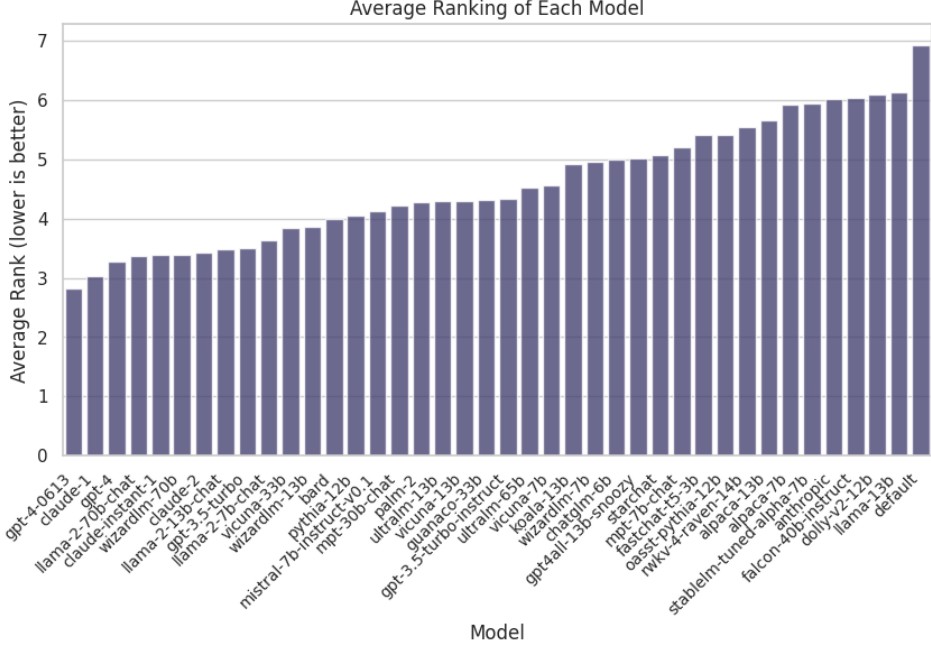

Figure 10: GPT-4-0613 (with system prompt for chat) had the highest average rank based on the prompt rubric, closely followed by Claude 1, GPT-4 (no system prompt), Llama-2-70b-chat, and Claude-Instant-1.

We extract the prompt intention classification (good natured or not good natured) from the rating response. From this we construct model performance when the prompt is good natured (in which helpfulness and harmlessness is expected) and when the prompt is not good natured (in which mainly harmlessness is expected). Note that these are GPT-4's estimation of the intention of the prompt, and are not necessarily fully accurate.

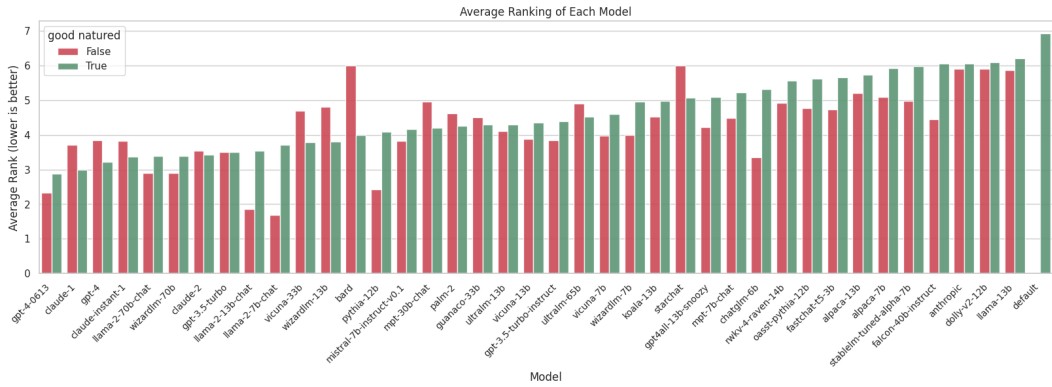

Figure 11: When considering only good natured user prompts, GPT-4-0613 had the highest average rank, based on the prompt rubric, closely followed by Claude 1, GPT-4 (no system prompt), Claude-Instant-1, and Llama-2-70b-chat.

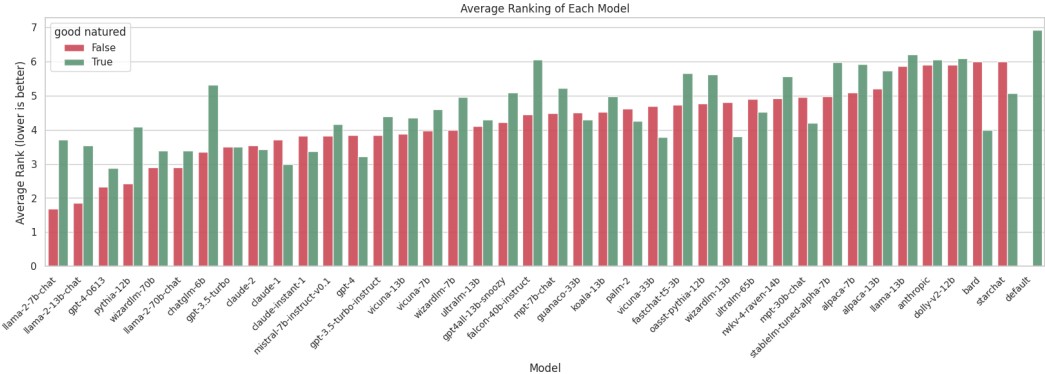

Figure 12: (Same graph as above, with x-axis sorted on when good natured is false) When considering only bad natured user prompts, Llama-2-7b-chat had the highest average rank, based on the prompt rubric, followed by Llama-2-13b-chat, GPT-4-0613, Pythia-12b, and Wizardlm-70b

The robust performance exhibited by Llama-2 in maintaining harmlessness can be attributed to its extensive safety alignment that counteracts prompt-level adversarial attacks (Chao et al., 2023). Prior research illustrates that Llama2 models possess impressive resilience when facing jailbreak prompts (Chao et al., 2023); (Zheng et al., 2023a). Additionally, contrary to expectations, smaller models have been empirically shown to outperform larger ones when assessed solely on harmlessness (Kundu et al., 2023). Our findings corroborate this notion, signifying that, in comparison to its counterparts, Llama-2-7b-chat consistently displayed the utmost harmlessness in response to ill-intentioned prompts. This was followed closely by Llama-2-13b-chat and Llama-2-70b-chat.

We also provide the heatmap for the pairwise average ranking difference of all models in our Nectar dataset in Figure 13, and the pairwise winrates in Figure 14.

# B  Reward Model Evaluation

## B.1  Evaluation Metrics

### B.1.1  Human Preference

For the human preference measurement, we first subsampled $10,000$ single-turn human-rated response pairs from Lmsys's ChatBot Arena Conversations dataset (Zheng et al., 2023b). We have each reward model return a score for each prompt response in the dataset. For a given rated response pair, if the reward model's score for the human-chosen winning response is greater than the score for the losing response, that response pair is considered classified correctly. The accuracies and losses for each model are detailed in Table 3 and Table 4.

### B.1.2  Truth

In order to understand if each reward model will value true statements over false statements, we utilize true and false response pairs from the Truthful QA dataset (Lin et al., 2021). For each prompt in the dataset, we sample three answers: the best truthful answer (Best), a true answer (True), and a false answer (False). We evaluate reward model output scores for each of these answers and average accuracy/loss across the three possible pairwise combinations where the overall ranking order is (Let $r$ be the reward model) $r(\text{Best}) >= r(\text{True}) > r(\text{False})$.

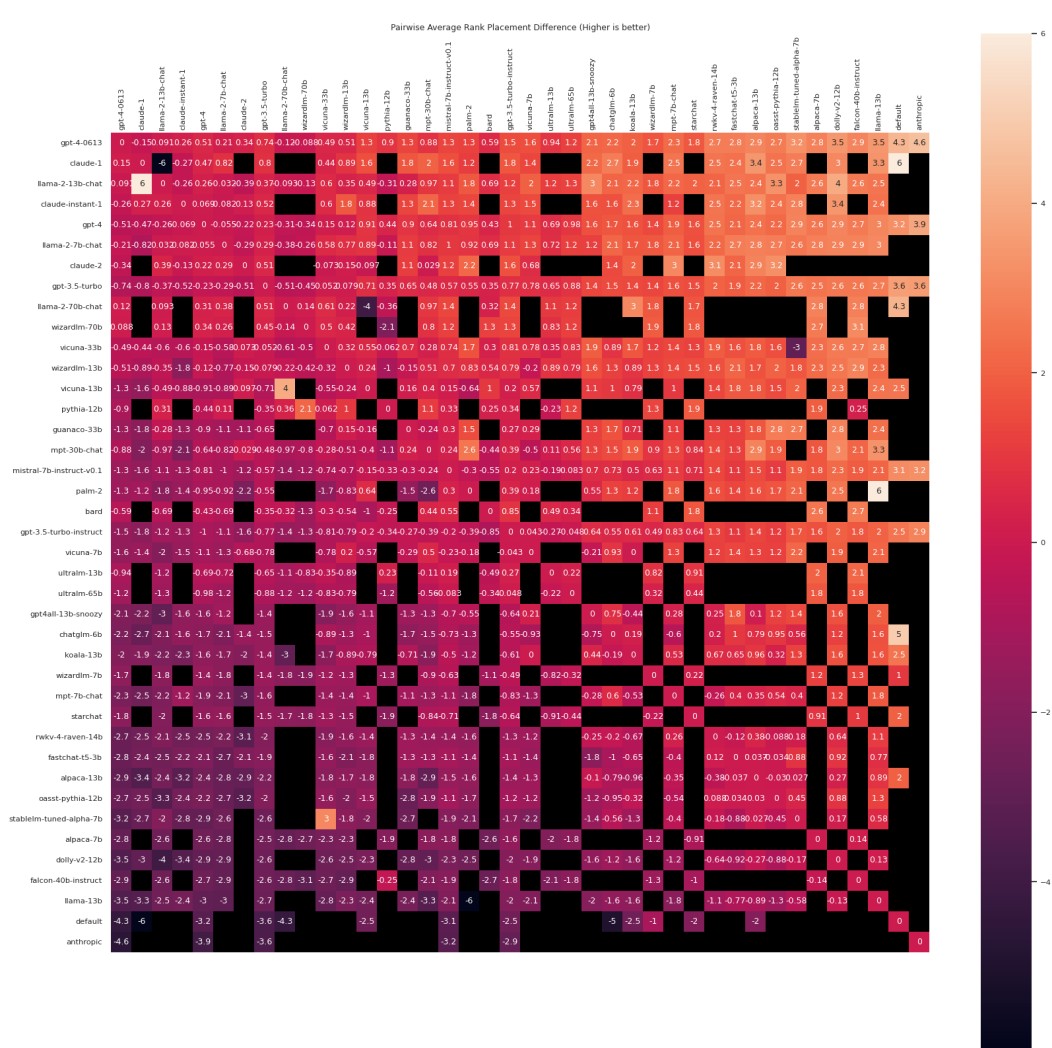

Figure 13: Heatmap of pairwise ranking difference for all models in Nectar.

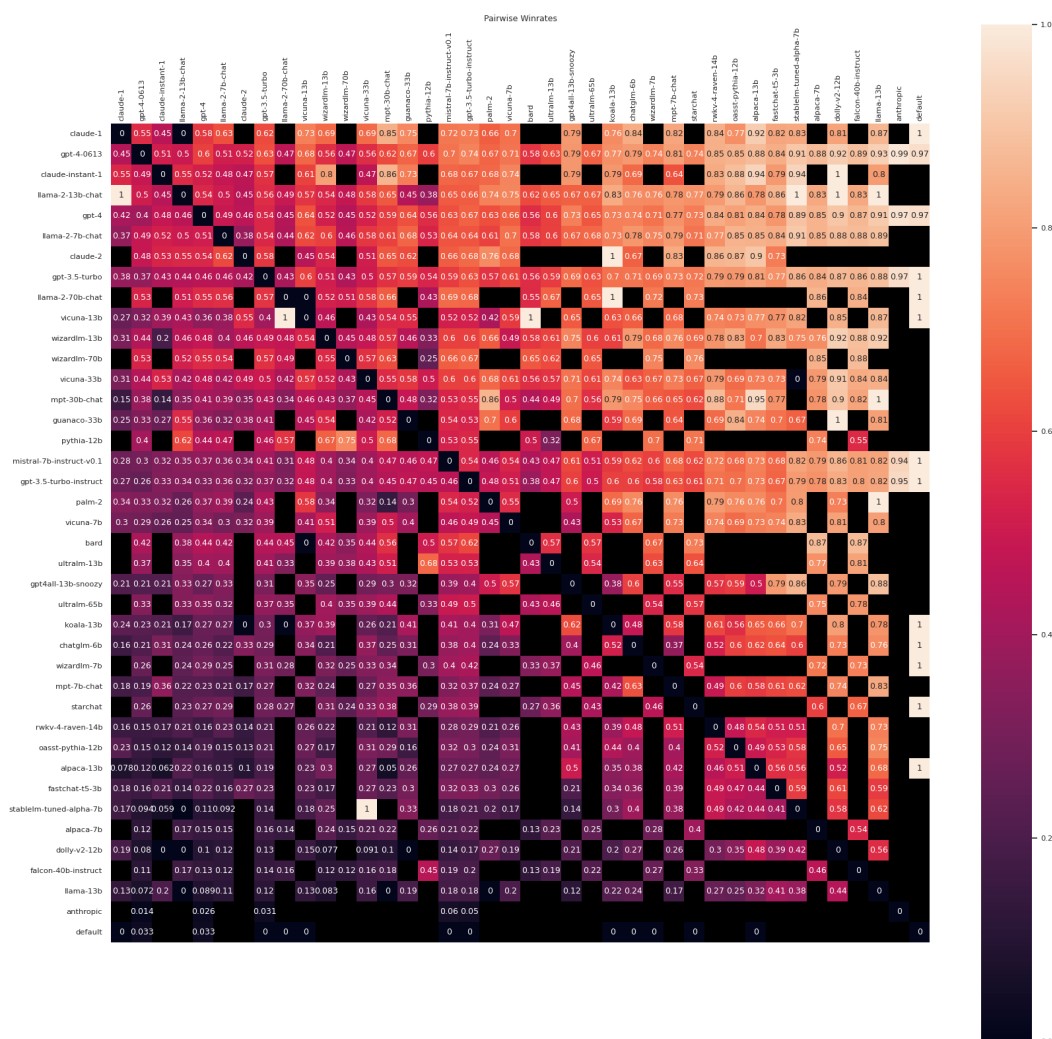

Figure 14: Heatmap of pairwise winrates for all models in Nectar.

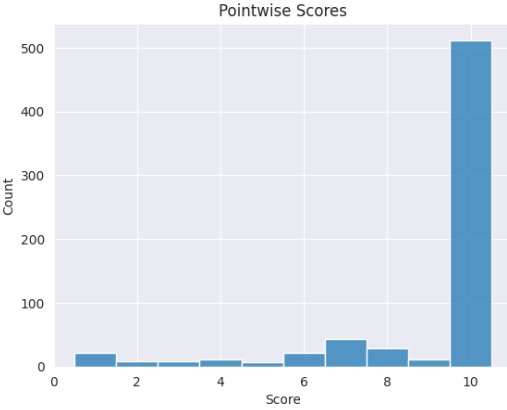

Figure 15: Pointwise scoring methods struggle to distinguish the differences between response choices and overwhelmingly give high scores.

### B.1.3   Safety

We measure safety by evaluating on the PKU-SafeRLHF dataset from Ji et al. (2023) sub-sampled down to 1000 response pairs where one response is safe and the other response is unsafe. Reward model scores are generated for each of the responses with the expectation that the the safe response score is higher than the unsafe response score. See Table 3 for more detail and Table 4 for losses.

### B.1.4   Verbosity

Empirical analysis of the RL fine-tuned language model shows an occasional undesirable tendency towards overly verbose responses to simple prompts. For example when prompted with "1+1", the model responded with:

> *Here are some important tips to consider when using the OpenAI GPT-4 model:*
>
> • *Understand the capabilities and limitations of the model: GPT-4 is a powerful language model, but it's important to recognize its limitations. While it's great at natural language understanding and generation, it may not always provide accurate or reliable information.*
>
> • *Be mindful of biases: ...* [response continues for another 700 words]

We created a small handcrafted dataset of 29 response pairs, where one response was overly verbose, and the other was not. Verbose response were crafted through finding prompts that yielded over verbose responses, handcrafting verbose response by introducing repetition, and through finding existing examples. A generated reward scoring pair is correct if the less verbose response receives the higher score.

In this task, all three reward models trained struggled, with accuracy rates of 0.167, 0.133, and 0.100 (Table 3). The reward model seems to often value extremely long responses and give lower scores to very simple but concise answers. In the above example where "1+1" is prompted, the long $700+$ word response was given a score of $-0.635$ while the simple but correct answer of "2" was given a score of $-4.313$. An additional example can be found in the Appendix B.2.

### B.1.5   GPT-4 Verbosity Preference

This section aims to delve into the implications of the verbosity preference noticed in the trained reward model. The investigation involves examining whether this verbosity preference materializes as a byproduct of GPT-4's inherent propensity while curating the rankings for Nectar. We run the same 29 evaluation pairs, comprising of both verbose and non-verbose responses, through the GPT-4 ranking procedure, which was adjusted for pairwise ranking. With the original rubric GPT-4 picks the non-verbose (correct) response with an accuracy rate of only 0.433. Augmenting the rubric to detail that response should not be "overly verbose" yields a slightly improve accuracy of 0.500. However, in both cases, GPT-4 is unable to correctly distinguish when verbosity should be less preferred. These experiments suggest that the verbosity bias noticed in the reward models is possibly a manifestation of GPT-4's inherent rating bias. An example of this verbosity preference is show below in the Appendix B.3.

## B.2 Additional Verbosity Example

> **Prompt**
>
> *Which of these fruits is a citrus? Give only the correct letter: A, B, C, or D.*
>
> *(A) Pear*
> *(B) Apple*
> *(C) Orange*
> *(C) Cherry*

> **Verbose Response**
>
> *Here are the steps to solve this problem:*
>
> *Only one of the given options is a citrus fruit. Citrus fruits belong to the Rutaceae family and include oranges, lemons, limes, and grapefruits. Pears, apples, and cherries, on the other hand, belong to different fruit families.*
>
> *Therefore, the correct option is:*
> *(C) Orange*
>
> *The correct answer is: C*

> **Non-Verbose Response**
>
> *C*

This prompt specifically instructs the model to only give the correct letter in the response. The verbose response ignores this instruction and gives additional information. In this case, the verbose response received a higher reward score of $-2.112$ and the non-verbose response received a lower reward score of $-4.065$.

## B.3 GPT-4 Verbosity Preference Example

> **Prompt**
>
> hi!

> **Response A (Verbose)**
>
> Hello! I'm here to assist you with any questions or tasks you may have.
>
> Here are some tips to keep in mind while using this AI:
>
> • Be clear and concise with your questions or requests.
> • Feel free to ask follow-up questions if you need more information or clarification.
> • Keep in mind that the AI is a powerful tool, but it may not always provide the exact answer you're looking for, especially in complex or nuanced situations.
> • While the AI can provide general advice, it's not a substitute for professional guidance from experts in specific fields.
> • Be respectful and courteous in your interactions with the AI, as it's designed to help you in a friendly and helpful manner.
> • If you have any specific questions or tasks, feel free to ask, and I'll do my best to assist you.

> **Response B (Non-verbose)**
>
> Hello! How can I help you today?

> **GPT-4 Rating**
>
> Both responses are polite, respectful, and adhere to moral and ethical standards. However, Model A's response is more detailed and provides useful information about how to interact with the AI, which makes it more helpful in this context.
> Ranking: [['A', 'B']]

In this judging context, despite being told in the system prompt that responses should not be overly verbose, GPT-4 still ranks the exceptionally verbose response over the simple response. It may be difficult to induce GPT-4 to rank verbose responses lower in conversational contexts were verbosity is actually less preferred.

