# OpenReview forum: "Starling-7B: Improving Helpfulness and Harmlessness with RLAIF"
_colmweb.org/COLM/2024/Conference — COLM_

### Official Review · Reviewer_dC9a · 2024-05-06

**Rating:** 8
**Confidence:** 3
**Ethics Flag:** 1

**Summary:**

* The authors contribute to open source:
    * The Nectar dataset, a preference dataset comprised of 3.8M pairs, significantly larger than many other open source preference datasets like Ultrafeedback and Anthropic-HH
    * 7B and 34B reward models trained on this dataset
    * Two 7B models finetuned with PPO using the 7B & 34B RMs respectively from Openchat-3.5 (“Starling-LM-7B”) which achieves strong results on MT-Bench, AlpacaEval, and human evaluations
* The authors detail various tips and tricks for collecting the Nectar dataset (for example, ways to reduce the positional bias problem when using a model-as-judge)
* Three different methods for training a reward model are compared, and the authors find applies the maximum likelihood estimator under Plackett-Luce model gives the best results
* The authors also detail various tips and tricks for improving stability and training robustness of PPO such as:
    * Shifting the reward mean for length control, to prevent model from becoming too verbose
    * Pretraining the critic model to stabilize early stages of training
    * Full parameter tuning, vs. unfreezing only top layers

**Questions To Authors:**

- Formatting nits: table reference on pg. 2 is broken

**Reasons To Accept:**

* The authors contribute a large and high quality preference ranking dataset, which is an order of magnitude larger than some comparable datasets like Ultrafeedback and Anthropic-HH, which will be very impactful for studying RLHF in open source and academic communities
* The authors also contribute several reward models trained on this dataset and two strong chat models trained with these reward models
* The authors provide many details on how to finetune a model with PPO; these types of tips and tricks are often left unpublished so it is great to see them detailed here

**Reasons To Reject:**

While the paper provides a detailed recipe for finetuning a model with PPO, the authors provide few comparisons to relevant baselines, for example (1) how does a reward model trained on the Nectar dataset compare to a reward model trained on some of the other open source datasets? (what exactly are we getting from the Nectar dataset over other datasets?) (2) how does the benchmark performance of the chat models compare to other similarly sized models?

---

> ### Author Rebuttal · Authors · 2024-05-31
>
> Thank you for your feedback. We appreciate your recognition of the importance of the problem our paper addresses
>
> >While the paper provides a detailed recipe for finetuning a model with PPO, the authors provide few comparisons to relevant baselines, for example (1) how does a reward model trained on the Nectar dataset compare to a reward model trained on some of the other open source datasets? (what exactly are we getting from the Nectar dataset over other datasets?) (2) how does the benchmark performance of the chat models compare to other similarly sized models?
>
>
> Thank you for your question. We will update our manuscript accordingly. In particular, we will include a comparison of the reward model with other existing reward models in Table 5. The Nectar dataset offers a larger amount of equivalent pairwise comparisons with higher quality generations, which appears to be the key for developing a high-quality reward model. Additionally, our Starling model is currently the best Mistral-7B-based model based on anonymous human evaluations conducted on Chatbot Arena.

---

### Official Review · Reviewer_viaJ · 2024-05-07

**Rating:** 6
**Confidence:** 4
**Ethics Flag:** 1

**Summary:**

Paper proposes Nectar, a high quality open-source preference dataset using GPT-4 to rank responses. This datasets was made based on asking model to consider all pairings before reaching ranking. The dataset is around 180k prompts and with 7 responses each, reaches 3.8 million pairwise comparisons. These prompts are collected from LMSys-1M, Anthropic-HH , Ultrafeedback and ShareGPT. Compared to existing general datasets (excluding StackOverflow which is coding specific), the number of pairwise comparisons is an order of magnitude higher.

Responses are ranked (ordinally) on both helpfulness and harmlessness. Responses are by models like gpt-4, gpt-3.5, Mistral 7b and llama2 7b. Using this dataset, they train Starling 7B, which is the best 7B model on Chat Arena. This model increase MT Bench from 7.81 to 8.09 on MT Bench and 88.51 to 91.99 on 91.99 and Chat Arena ELO from 1072 to 1087. When trained with a larger reward model, the Chat Arena Elo increase further to 1118.

**Questions To Authors:**

1.	The choice of base models used with the Reward model is weird. Why did you choose to use llama 2 7b and then Yi 34B (as opposed to 2 models in the same family, which is available)? Specifically using Llama 2 7b might mean that the reward model can only be used to train models from the llama family (when the generally better-viewed Mistral 7b with Apache 2.0 license is available).
2.	Please take care to correct grammatical errors (likey should be likely) and citation errors such as TRLX (Havrilla et al.) should be (Havrilla et al., 2023). These are plenty in the paper as it seems rushed and I trust the authors to correct them later on.

**Reasons To Accept:**

1.	The approach to rate responses is well designed to solve the positional bias issue which is well illustrated in Figure 2. Specifically, they do pairwise ranking and summarize into listwise ranking – all in a single prompt.
2.	The reward model training approach is reasonable to allow K-ary comparisons rather than decomposing it into 7C2 pairwise comparisons.
3.	The trained reward models and policy models are evaluated on a good range of evaluation metrics and show promising results on these benchmarks.
4.	The design of adding a slightly positive reward to allocate some advantage to EOS token to prevent length explosion is reasonable. Similarly, pretraining the critic is an interesting finding that stabilizes training.
5.	Monitoring Verbosity and IFEval as a strategy for checkpoint selection is reasonable.

**Reasons To Reject:**

1.	In Figure 6 right-most plot, while most of the positional bias from position 1 and 3-6 are eliminated, there is a curious higher win-rate for position 0 and lower win rate for position 2. Can the authors explain why this is the case?
2.	Can the authors justify why Ordinal ranks were used rather than Cardinal as used in Ultrafeedback and HelpSteer dataset (see Wang et al. (2023) “HelpSteer: Multi-attribute Helpfulness Dataset for SteerLM” http://arxiv.org/abs/2311.09528)? Cardinal scores (say between 1 to 10) can strictly convey more information (as we can construct ordinal ranks from cardinal scores but not the other way round) and does not require ties breaking as required in the ordinal approach. More importantly, cardinal scores can convey how much better a response is compared to other responses say [10, 1, 1,1,1,1,1] is pretty different from [10,9,8,8,7,7,7] even though they may have the same ordinal ranks.
3.	According to Figure 4, the top 4 models used (~180k each) are all from OpenAI (gpt4 and gpt3.5 variants) followed by mistral-7b and anthropic. This means that for any given prompt, there are two responses from gpt-4 and another two from gpt-3.5. Given the judge is also gpt-4, it is very likely to choose gpt-4 responses as the top rated response. Sharma et al. (2024) “A Critical Evaluation of AI Feedback for Aligning Large Language Models” https://arxiv.org/pdf/2402.12366 shows that much of gains from AI Feedback by GPT-4 can in fact be attributed to distillation from GPT-4. Relating back to this paper, this paper is missing a key ablation of simply doing SFT on top of the GPT-4 response to each prompt.
4.	Using MT Bench to select policy model checkpoint and also reporting as an policy model metric risks overfitting to this benchmark.
5.	Authors do not mention the specific LR and KL penalty used, which makes the RM and PPO experiments hard to reproduce.
6.	The authors mention using 10-20 challenging prompts for internal human evaluation but do not share what these prompts are and specifically how the evaluation is done. Similarly, they do not mention specific details for varies approach to checkpoint selection (such as what is the length threshold they used).

---

> ### Author Rebuttal · Authors · 2024-05-31
>
> We appreciate the detailed feedback and will endeavor to address your concerns.
> >In Figure 6 right-most plot…
>
> We are choosing one fixed ordering that helps mitigate the positional bias. However, fixed ordering might lead to some disadvantage for responses in position 2. In practice, one may use random ordering to mitigate this.
> >Can the authors justify why Ordinal ranks were used rather than Cardinal…
>
> Since most of our responses are generated from strong models, the scores for responses will heavily concentrate around 9-10 (out of 10), which essentially provides much less signal than pairwise or multi-wise comparison. We have also tried a score range 0-100. However, we still observe a high concentration around close to 100, which weakens the information we can get.
>
> >According to Figure 4…
>
> Since OpenChat-3.5 has already undergone extensive GPT-4 distillation during the SFT phase with millions of prompts, much larger than our scale, additional SFT will result in no improvement or even hurt performance due to overfitting. However, RLHF can still improve the model’s ability even if SFT has already saturated.
> >Using MT Bench to select policy model checkpoint … risks overfitting to this benchmark.
>
> Our goal is not to overfit MT Bench, but to excel in Chatbot Arena, which is purely based on human judgment and hard to overfit (initial evaluation requires around 2 weeks). MT Bench is only for filtering out most of the checkpoints that were suboptimal before we spent human resources to evaluate them thoroughly.
> >Authors do not mention the specific LR and KL penalty…
>
> We will report the LR and KL penalty in our final revision.
>
> >The authors mention using 10-20 challenging prompts but do not share what these prompts…
>
> For the IFeval-based checkpoint filtering strategy, we manually define the threshold. In Starling 7B beta training, we set this threshold to be 0.05 regression compared with the SFT model. For the human evaluation prompts, we will also release them in our final revision.
> >The choice of base models used with the Reward model is weird…
>
> We observed a lower loss and better convergence compared to training Mistral 7B as a reward model. It could be the difference in their model architecture or coverage of their pretrain data. We think this might raise interesting research questions which require further studies.
> >Please take care to correct grammatical errors…
>
> We will correct them in our revision, and we appreciate the reviewer for pointing them out!

---

> > ### Comment · Reviewer_viaJ · 2024-06-05
> > **Response to Rebuttal**
> >
> > Thanks for the rebuttal, which answers some of my questions particular regarding why Ordinal ranks instead of Cardinal ranks, why the authors didn't include a GPT-4 SFT baseline and the use of MT Bench as checkpoint selection.
> >
> > Please include these updates into the revised paper.
> >
> > On the other hand, I still find that the amount of detail that the authors share in the paper and the rebuttal (excluding what they promise since some of it can be trivially answered in the rebuttal) is not sufficient for good replication of the training procedure. Especially since RLHF is known to be unstable, this makes replication challenging.
> >
> > It is also worrying that RM training on the Mistral 7B RM does not work well - authors did not bring this up initially and even after the rebuttal, offered very little data regarding how the Mistral 7B RM does (e.g. Rewardbench) , even though theoretically, a strong preference modeling dataset with millions of preference pairs should be able to train Mistral 7B as well as Llama2 7B. This potentially means that the preference dataset they introduce might only work on specific models, which is extremely surprising and should be discussed with some detail in the paper.
> >
> > As impressive as the results are (that I mentioned in the Reasons to Accept), I find the lack of training details concerning. Therefore I will keep my scores the same.

---

> > > ### Author Response · Authors · 2024-06-06
> > >
> > > >Please include these updates into the revised paper.
> > >
> > > Yes, we will definitely include all the changes into the revised paper.
> > >
> > > >On the other hand, I still find that the amount of detail that the authors share in the paper and the rebuttal (excluding what they promise since some of it can be trivially answered in the rebuttal) is not sufficient for good replication of the training procedure. Especially since RLHF is known to be unstable, this makes replication challenging.
> > >
> > > We appreciate your concern about replicability. We will release the full training code along with detailed documentation in the final revision. This will include the exact hyperparameters used, which should greatly facilitate replication efforts. We acknowledge that RL training can be noisy, but we believe providing these details will help others reproduce our results more consistently.
> > >
> > > >It is also worrying that RM training on the Mistral 7B RM does not work well - authors did not bring this up initially and even after the rebuttal, offered very little data regarding how the Mistral 7B RM does (e.g. Rewardbench) , even though theoretically, a strong preference modeling dataset with millions of preference pairs should be able to train Mistral 7B as well as Llama2 7B. This potentially means that the preference dataset they introduce might only work on specific models, which is extremely surprising and should be discussed with some detail in the paper.
> > >
> > > While it's surprising that our dataset did not yield similar results on Mistral 7B, we want to highlight that training Yi 34B as a reward model using our dataset led to excellent performance, surpassing even some larger models. This demonstrates the high quality of our preference data. We will explore potential reasons for the discrepancy between Mistral 7B and other models, and discuss the implications in the revised paper.

---

### Official Review · Reviewer_nUFE · 2024-05-10

**Rating:** 7
**Confidence:** 4
**Ethics Flag:** 1

**Summary:**

Quality: The authors have developed an approach to utilize GPT-4 to create a preference dataset, Nectar, and train reward models. Finally the authors train a chat model with PPO using the reward models. Using this pipeline, they achieve state-of-the-art performance within the 7B scale models. So the quality is ok.

Clarity: The paper is easy to follow.

Originality: The paper combines a few existing things together, it is not particularly novel. The most novel part is they do global ranking so that the reward models have more samples to learn from

Significance: This work makes a great contribution to the open-source community. The reward model achieves pretty good performance on rewardbench, and the model is publicly released.

**Questions To Authors:**

1. I would recommend authors remove ShareGPT from their dataset. ShareGPT is not published and never obtains users' consent to share and release their data. It is unethical to redistribute data without proper permission. To be clear, this is a suggestion and not part of my paper evaluation.
2. It is not clear to me why related work is part of introduction?
3. From Figure 1, it looks like the only significant improvement of the model is on the reasoning category. Does the author have an explanation for why this is the case?
4. Figure 13 and 14 are not for readers to understand.

**Reasons To Accept:**

1. Both the reward model and the chat model achieve state-of-the-art performance.
2. The data and models can benefit future research in the field.
3. The proposed ranking method seems simple and works well.

**Reasons To Reject:**

1. Lack of novelty, mostly prompt engineering to get good ranking information.
2. The paper presentation can be improved. Some of the figures are not particularly readable, and there are small typos throughout the paper.

---

> ### Author Rebuttal · Authors · 2024-05-31
>
> Thank you for your feedback. We deeply appreciate the opportunity to enhance the clarity of our work.
>
> >Lack of novelty, mostly prompt engineering to get good ranking information.
>
> We would like to emphasize that our main contribution is to demonstrate the possibility of providing a comprehensive recipe to reproduce a strong and deployable model. This includes data creation, data curation, and detailed algorithm implementation, all of which are crucial for achieving great results. We identify and propose simple prompting techniques to facilitate our goal, and we are the first to seriously attempt employing the less-explored K-wise reward training loss. Additionally, we propose several important implementation details that are extremely important for stable PPO training, such as shifting the reward for length control and pre-training the critic function. We believe this comprehensive recipe provides a significant contribution not only in engineering but also in research.
> >The paper presentation can be improved. Some of the figures are not particularly readable, and there are small typos throughout the paper.
>
> We will correct the typos and improve the quality of the figures in our revision. Thank you for the suggestion.
> >I would recommend authors remove ShareGPT from their dataset. ShareGPT is not published…
>
> Thank you for raising this concern. We will take it into consideration when updating our dataset.
> >It is not clear to me why related work is part of introduction?
>
> Thanks. We will move the related work section to be independent of the introduction.
> >From Figure 1, it looks like the only significant improvement of the model is on the reasoning category. Does the author have an explanation for why this is the case?
>
> The original OpenChat 3.5 model has already achieved very strong performance in categories like writing and humanities, while still significantly lagging behind closed-source models like Claude or GPT-4 in STEM-related domains such as coding and reasoning. This discrepancy provides more room for improvement in these underperforming domains. Furthermore, our dataset contains a reasonable amount of coding and reasoning-related questions, which may explain the observed improvements.
> >Figure 13 and 14 are not for readers to understand.
>
> We will add detailed explanations about Figures 13 and 14 in the Appendix.
>
> We appreciate your feedback and hope that these clarifications will make our revised draft more comprehensible and informative.

---

> > ### Comment · Reviewer_nUFE · 2024-06-05
> > **Thanks for the rebuttal**
> >
> > Thanks for the rebuttal. Please do take these promised changes in the revision. Otherwise, I'd love to see this paper accepted.

---

### Official Review · Reviewer_Xpnc · 2024-05-11

**Rating:** 7
**Confidence:** 3
**Ethics Flag:** 2

**Summary:**

This paper presents not only a chatbot which is the best on the chatbot area, but also releases the training dataset, and explains implementation details.

**Ethics Concerns Details:**

the paper mentions distilling from GTP-* output. I am not sure if that is allowed under the Terms of Service?

**Questions To Authors:**

small comments:

- there is a messed up reference:
"We compare the Nectar dataset with existing preference dataset (Cui et al., 2023; Bai et al., 2022a; Ethayarajh et al., 2022) in Table ??.'

- figure 1 is blurry.

Typos:
-  "We would like to When K = 2, "
- Similar to the default methodology of training reward model  -->  training a reward model, or training reward models

**Reasons To Accept:**

- The analysis on positional bias has potential to be useful to other work.
- The paper in general does a good job of explaining implementation details, which are often glossed over.
- The model is the best-performing 7B chat model on Chatbot Arena (at time of publication) publishing the details about it will be of use to the community as a while.

**Reasons To Reject:**

- There are quite a few limitations in using GPT as an evaluator, though the authors acknowledge those.
- While this is a useful open source tool, data release, and hyper parameter optimization, there is more limited research contribution.

---

> ### Author Rebuttal · Authors · 2024-05-31
>
> Thank you for your insightful comments. We appreciate your recognition of the importance of the problem that our paper addresses. We value the opportunity to further illustrate the key aspects of our research.
>
> >There are quite a few limitations in using GPT as an evaluator, though the authors acknowledge those.
>
> Yes, we not only address the positional bias in GPT4 evaluation but also provide a simple prompting technique to mitigate the bias. We believe our methodology could benefit future research.
> >While this is a useful open source tool, data release, and hyper parameter optimization, there is more limited research contribution.
>
> We would like to emphasize that our main contribution is to demonstrate the possibility of providing a comprehensive recipe to reproduce a strong and deployable model, this includes data creation, data curation, algorithm implementation details, which are all very important to obtain great results. We identify and propose simple prompting techniques to facilitate our goal, and we are the first serious attempt to employ the less explored K wise reward training loss. We also propose several important implementation details that’s extremely important for stable PPO training, e.g., shifting the reward for length control and pre-training the critic function. We believe this comprehensive recipe already provides a significant contribution not only in engineering but in research.
>
> We hope these clarifications address your concerns and provide a deeper understanding of our work. We are grateful for your engagement and look forward to further discussions.

---

### Decision · Program_Chairs · 2024-07-10

**Decision:**

Accept

**Comment:**

This paper presents a recipe for training a model with RLAIF. The final Starling-7B model achieves high performance on ChatbotArena, among other benchmarks.  While there are some methodological criticisms (e.g., data skewed towards GPT-4, by viaJ), the reviewers on balance find the Nectar dataset, its reward models, and the final LLMs helpful. Some of the details like the analysis of positional bias are likely to be useful in other work.